# Survival Analysis for Idiopathic Pulmonary Fibrosis using CT Images and Incomplete Clinical Data

**Ahmed H. Shahin**[1,2]                                        AHMED.SHAHIN.19@UCL.AC.UK
**Joseph Jacob**[2]                                                      J.JACOB@UCL.AC.UK
**Daniel C. Alexander**[2]                                    D.ALEXANDER@UCL.AC.UK
**David Barber**[1]                                              DAVID.BARBER@UCL.AC.UK

[1] *Department of Computer Science, University College London, UK*

[2] *Centre for Medical Image Computing, University College London, UK*

**Editors:** Under Review for MIDL 2022

## Abstract

Idiopathic Pulmonary Fibrosis (IPF) is an inexorably progressive fibrotic lung disease with a variable and unpredictable rate of progression. CT scans of the lungs inform clinical assessment of IPF patients and contain pertinent information related to disease progression. In this work, we propose a multi-modal method that uses neural networks and memory banks to predict the survival of IPF patients using *clinical and imaging data*. The majority of clinical IPF patient records have missing data (e.g. missing lung function tests). To this end, we propose a probabilistic model that captures the dependencies between the observed clinical variables and imputes missing ones. This principled approach to missing data imputation can be naturally combined with a deep survival analysis model. We show that the proposed framework yields significantly better survival analysis results than baselines in terms of concordance index and integrated Brier score. Our work also provides insights into novel image-based biomarkers that are linked to mortality.

**Keywords:** survival analysis, IPF, interstitial lung diseases, neural networks

## 1. Introduction

Idiopathic Pulmonary Fibrosis (IPF) is the most common and deadly fibrotic lung disease with a median survival rate ranging from 2.5 to 3.5 years (Katzenstein and Myers, 1998; Vancheri, 2013). IPF is characterized by stiffening and scarring (fibrosis) of the lung tissue that leads to shortness of breath and progressive reductions in lung volume. Spirometric evaluation including measurements of Forced Expiratory Volume in the first second (FEV1) and Forced Vital Capacity (FVC) captures alterations in lung volume that occur in IPF. One of the main challenges with IPF is the unpredictable and highly-variable disease progression seen across individuals. IPF progression is described by worsening respiratory symptoms, lung function decline, progressive fibrosis on Computed Tomography (CT) imaging, or death. The majority of patients suffer from progressive lung function decline. While FVC is used to track IPF progression (Jegal et al., 2005), mortality is considered the most reliable objective endpoint (Raghu et al., 2012). It can be interpreted in any of the following forms: all-cause mortality, respiratory-related mortality, or IPF-related mortality. The most clinically relevant expression of mortality is all-cause mortality (King et al., 2014), which is used in this paper to model disease progression in IPF patients.

---

. The source code is publicly available at: https://github.com/ahmedhshahin/IPFSurv

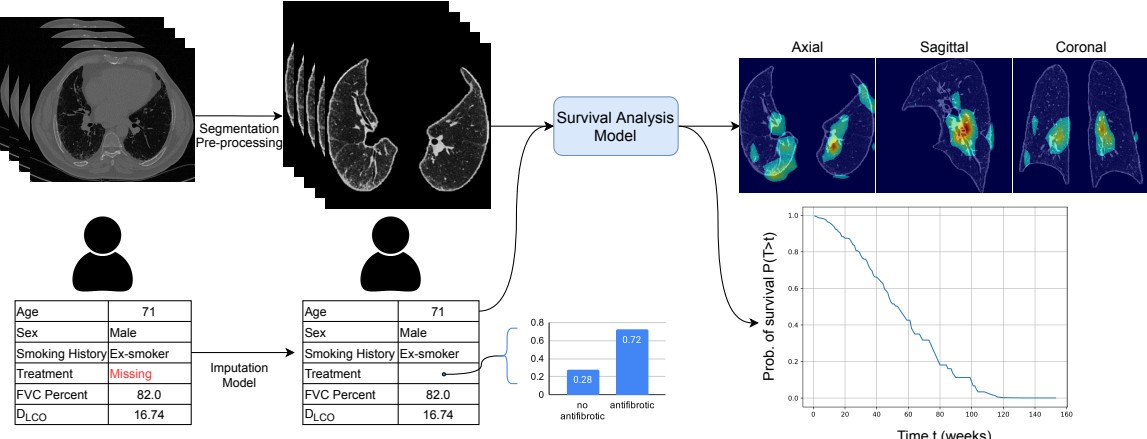

Figure 1: Overview of the proposed framework. During survival analysis training, we sample from the posterior distribution of missing values conditioned on the observed values (Section 2.1), while during testing we use the most probable value.

A related challenge is that clinical records associated with the CT scans contain missing clinical data, with more than 65% of our dataset containing at least one missing value. Consequently, training survival models on complete samples would drastically reduce the amount of training data and negatively impact the survival analysis model performance. We propose a fully-automated survival analysis framework to discriminate IPF patients according to their mortality risk, while being robust to missing clinical data. Our framework can be used to assess the mortality risk at any disease stage using clinical and imaging data.

## 2. Methods

Our work contains two main contributions. The first is a simple yet principled approach to dealing with missing values in clinical records. This allows us to train a subsequent deep network to predict patient survival time using both the patient's CT image and clinical record, with any missing clinical values sampled from the missing data model (see Figure 1). The second is a deep survival model supported with a memory bank to enable more efficient processing of 3D volumetric images.

### 2.1. Imputation of missing values

Missing data can be imputed in many ways, see for example (Barber, 2012; Stavseth et al., 2019). However, incautious handling can bias the model adversely. For example, imputing with zeros might lead to correlating a missing value with a poor prognosis due to the inability of patients in late stages to perform the lung function tests (Yi et al., 2019). Similarly, imputing with mean values (Donders et al., 2006) assumes all data attributes are independent, which is an invalid assumption in the case of IPF clinical features (see Appendix C). Taking dependency between attributes into account, Multiple Imputation by Chained Equations (MICE) (Azur et al., 2011) is an algorithm that iteratively performs supervised regression to

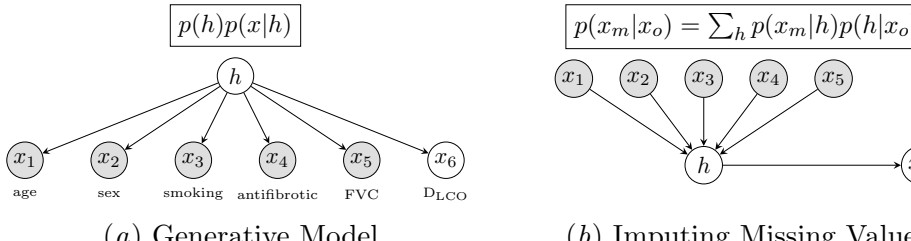

$(a)$ Generative Model $\qquad$ $(b)$ Imputing Missing Values

Figure 2: Model for imputing missing clinical values. $h$ represents the hidden state and $x$ is the patient clinical information. Black circles denote observed variables $x_o$ and white circles denote missing variables $x_m$ (D$_{\text{LCO}}$ in this example).

model missing data conditioned on observed data. HI-VAE (Nazábal et al., 2020) proposed learning a different likelihood function for each data type (e.g. continuous and discrete) and combining them in a variational auto-encoder model (Kingma and Welling, 2014).

We therefore introduce a simple latent variable model that is computationally efficient. To impute missing values, we assume the clinical features $x$ are modelled by independent categorical distributions, when conditioned on a hidden state $h$, see Figure 2. For patient $n \in \{1, \ldots, N\}$, the probability of clinical record $x^n$ under the model is therefore given by

$$p(x^n) = \sum_{h=1}^{H} p(h) \prod_{k=1}^{K} p(x_k^n|h) \tag{1}$$

where $p(h)$ denotes[1] a categorical distribution with state $h \in \{1, \ldots, H\}$; $K$ is the number of clinical features, and $p(x_k^n|h)$ is a categorical distribution. Writing each record in terms of observed and missing elements, $x = (x_o, x_m)$, the likelihood of record $x^n$ is given by

$$p(x^n) = \sum_{h} p(x_o^n|h)p(x_m^n|h)p(h) \tag{2}$$

where $p(x_o^n|h) = \prod_{i \in x_o^n} p(x_i|h)$ and $p(x_m^n|h) = \prod_{i \in x_m^n} p(x_i|h)$. To model continuous features, we convert them into discrete variables by equal-frequency binning.

The model has two sets of parameters, the hidden distribution $p(h)$ and the categorical distributions $p(x_i|h)$. The Expectation–Maximization (EM) algorithm (Dempster et al., 1977) is a convenient choice to learn these distributions. Note that the EM algorithm can make use of all training data, even those records which contain missing data, see Appendix B. After training the model parameters, the distribution of missing values is computed from

$$p(x_m|x_o) \propto \sum_{h} p(h)p(x_m, x_o|h) = \sum_{h} p(h)p(x_m|h)p(x_o|h) \tag{3}$$

It is then straightforward to calculate missing data statistics or draw samples as required.

---

1. Throughout we use the compact notation $p(h)$ to denote an (unnamed) random variable (associated with state $h$) being in state $h$ and similarly for conditional distributions. This obviates writing for example $p(\tilde{H} = h)$ for random variable $\tilde{H}$ in state $h$.

## 2.2. Deep Survival Analysis

Taylor Gonzalez and Maher (2016) used Cox regression (Cox, 1972) to predict mortality from the Gender Age Physiology index (GAP) and Composite Physiologic Index (CPI). Collard et al. (2003) adopted a similar approach and concluded that six-month changes in pulmonary function tests were predictive of mortality risk. However, CT scans of the lungs constitute an important part of the clinical assessment of IPF patients and contain pertinent information related to disease progression. It can also be shown that patients with similar clinical information may have different prognoses (Appendix A). Therefore, we investigate the performance of survival models that use both imaging and clinical data.

Other studies have used extracted features from CT to predict mortality. Jacob et al. (2017) compared between mortality prediction using features extracted by an expert radiologist (visual scoring) and features automatically extracted by CALIPER software (Computer-Aided Lung Informatics for Pathology Evaluation and Ratings) (Bartholmai et al., 2013). CALIPER quantifies the extent of specified radiological patterns of lung damage[2] seen on the CT scan. However, both the visual scoring and CALIPER approaches are unsupervised feature extraction methods in the sense that they are not designed to be maximally predictive of mortality. Visual scoring is also a time-consuming approach that requires clinical expertise and is prone to inter-observer variability.

We are therefore interested in estimating the time to death of a patient, based on their clinical and imaging data. We train an end-to-end neural network to extract imaging features that are maximally predictive of mortality. In survival analysis (Kleinbaum and Klein, 2010), one may not know whether some patients have died or just stopped visiting the hospital; the only available information about these patients is that they were alive until a specific date (date of censoring). Writing $T^*$ for the time of death, the hazard function $h(t)$ models the chance that a patient will die in an infinitesimal time interval $[t, t + \Delta t]$

$$h(t) = \lim_{\Delta t \to 0} \frac{p(t \leq T^* < t + \Delta t | T^* \geq t)}{\Delta t} \tag{4}$$

The most widely used model to learn from censored survival data is the Cox proportional hazards model (Cox, 1972). It models the hazard function $h(t|x)$ conditioned on the feature vector $x$, as follows

$$h(t|x) = h_0(t) \exp(g(x)) \tag{5}$$

Here $h_0(t)$ depends only on $t$ and $g(x)$ is a deep network that depends on the patient covariates $x$. The parameters of $g(x)$ are learned by minimizing the negative partial log-likelihood function (Cox, 1972). To do this, for each patient $n$ we define the risk set $R_n$ as all those patients that have not died before patient $n$ and define the relative death risk as

$$P(T_n^* = t_n | R_n) = \frac{h(t_n | x^n)}{\sum_{m \in R_n} h(t_m | x^m)} = \frac{\exp(g(x^n))}{\sum_{m \in R_n} \exp(g(x^m))} \tag{6}$$

The negative partial log-likelihood is then defined as the sum of $\log P(T_n^* = t_n | R_n)$ for all patients who died $n \in \mathcal{D}$

$$L = -\frac{1}{|\mathcal{D}|} \sum_{n \in \mathcal{D}} \left[ g(x^n) - \log \left( \sum_{m \in R_n} \exp(g(x^m)) \right) \right] \tag{7}$$

2. Ground glass opacity, reticulation, honeycombing, emphysema, pulmonary vessels volume, and others.

---

**Algorithm 1:** Pseudocode of survival analysis training in a PyTorch-like style

---

```
# delta: indicator function (1 if experienced the event and 0 if censored), shape Nx1
# times: event or censoring time, shape Nx1
# case_idx: an identifier of each sample in training set, shape Nx1
mbank[case_idx] = rand(N) # initialize memory bank as a dictionary with random values
for (batch_cases, img, clinical) in loader: # load a minibatch with n samples
    clinical = impute_missing(clinical) # sample from p(x_m|x_o) to impute any missing
        values
    pred = model(img,clinical) # get prediction using imaging and clinical data
    mbank[batch_cases] = pred.data # update values of the current batch in the memory
        bank
    loss = CoxLoss(mbank.values, delta, times) # calculate loss using the whole dataset
        by accessing predictions in mbank from previous iterations
    loss.backward() # calculate gradients
    update(model.params) # update model parameters
```

---

Minimizing $L$ with respect to the parameters of $g(x)$ using standard stochastic gradient descent based on selecting batches of patients (Kvamme et al., 2019) is problematic since:

- Eq(7) represents a ranking loss that compares between patients that died in the batch according to their predicted mortality risk. This requires large batch sizes for robust training; however, for high-resolution inputs (3D scans) we are limited by GPU memory to small batch sizes.

- For small batch sizes (usually less than 10) and a high censoring percentage, there will often be batches containing only censored patients. The loss, in this case, cannot be calculated and these batches will be ignored.

Inspired by the contrastive learning literature (He et al., 2020), we introduce a **memory bank** to store neural network predictions. This allows the loss in Eq(7) to be approximately calculated on the whole training set.

To calculate the hazard function, we use a Convolutional Neural Network (CNN) to encode each CT into a vector representation and append it to the clinical data (with any missing clinical data sampled from the posterior $p(x_m^n|x_o^n)$) to get a joint representation (Gadzicki et al., 2020). Given the limited number of clinical features in the dataset (six features), using a more complicated multi-modal learning approach might lead to overfitting. The function $g(x)$ is then obtained by a linear combination of the elements of the joint representation. The overall training process, including how to deal with missing clinical data, is explained in Algorithm 1.

For the CNN part, we use an adjusted 3D ResNet that has been shown to work well with medical imaging scans in Pölsterl et al. (2021). It consists of four residual blocks, each block includes two $3 \times 3 \times 3$ convolutional layers, with ReLU non-linearity and batch normalization. To optimize the parameters of the model, we minimize the loss in Eq(7) using Adam optimizer with initial learning rates of 0.01 and 0.03 for imaging-only and multi-modal models, respectively. We train the models for 100 epochs and shrink the initial learning rate by 10 after 30 epochs. Learning rates were chosen via random search based on the best predictive performance. We use the model provided by Hofmanninger et al. (2020) for lung segmentation. In the imputation model, the latent variable has 90 discrete states.

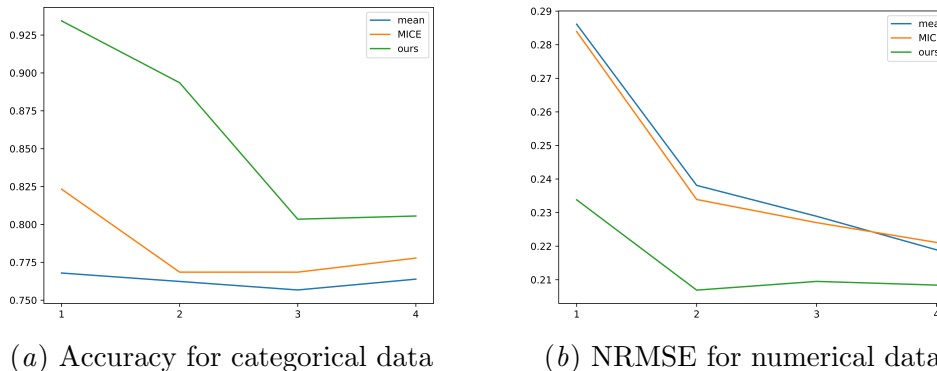

$(a)$ Accuracy for categorical data $\qquad$ $(b)$ NRMSE for numerical data

Figure 3: Results of imputation. X-axis: number of missing features per clinical record.

## 3. Experiments

In our experiments, we use the Open Source Imaging Consortium (OSIC)[3] dataset which contains lung CT scans as well as contemporaneous clinical data. We use six clinical features: age, sex, smoking history (never-smoked, ex-smoker, current smoker), antifibrotic treatment (yes/no), FVC percent, and Diffusion capacity of carbon monoxide ($D_{LCO}$). To ensure correspondence between imaging and clinical data, we only use patients who had their lung function tests within 3 months of the CT scan.

**Imputation of missing values** Our dataset comprises 761 patients with clinical data and only 182 patients have complete records. To validate the effectiveness of the model in Section 2.1, we use a test set of 72 patients with complete records and simulate missing values by randomly dropping 1 to 4 features from each test sample and imputing them using mean imputation, MICE (Azur et al., 2011), and our model. We assume that age and sex features are always observed as this information is available for all patients in the dataset, and is usually available in hospitals. To infer the missing values we take the expectation of Eq(3). In Figure 3, we show the average accuracy for categorical variables and normalized root mean square error (NRMSE) for numerical variables. For a vector of true values $y$ and a vector of imputed values by the model $\hat{y}$, NRMSE is defined as follows:

$$\text{NRMSE} = \frac{\sqrt{\frac{1}{|y|} \sum_i (y_i - \hat{y}_i)^2}}{max(y) - min(y)} \tag{8}$$

To account for randomness, we repeat experiments 5 times and report the average accuracy and NRMSE. These results demonstrate the superior performance of the proposed model compared to mean and MICE imputation. In Appendix C, we show the correlation between different clinical features suggesting that the performance improvement of our model comes from modelling the correlation between features.

**Survival analysis results** We use 446 IPF patients in this set of experiments. Each patient has a volumetric CT scan with slice thickness $\leq$ 2.0 mm and contemporaneous clinical data. We do five-fold cross-validation and use the standard Cox model using clinical

---

3. https://www.osicild.org

Table 1: Results of five-fold cross-validation for survival analysis

| Model | Memory Bank | IPCW C-Index | IBS |
|---|---|---|---|
| Cox (clinical data) | N/A | 63±19.23 | 0.33±0.1 |
| CNN (imaging data) | ✗ | 75.86±6.88 | 0.32±0.33 |
| | ✓ | **77.68±4.51** | 0.24±0.12 |
| CNN (both) | ✗ | 69.73±23.04 | 0.4±0.3 |
| | ✓ | 71.28±16.95 | **0.18±0.09** |

data as our baseline. We compare the baseline to models that use imaging only as input and models that use both clinical and imaging data. To evaluate the model discrimination, i.e. the ability to discriminate between patients according to their risk of death, we use a modified version of the concordance index (Harrell et al., 1996) based on the inverse probability of censoring weights (IPCW C-Index) (Uno et al., 2011). To evaluate the model calibration, i.e. the ability to precisely predict the time of death, we use the Integrated Brier Score (IBS) (Graf et al., 1999). For a survival probability $\hat{p}^t$ at time $t$, Brier score is defined as $BS(t) = \frac{1}{N^*} \sum_i (\mathbb{1}_{(T^*>t)} - \hat{p}_i^t)^2$ where $N^*$ is the set of uncensored patients or patients who have censoring dates later than $t$. IBS is obtained by integrating BS over a time interval, the interval that spans the test set in our experiments.

Table 1 shows that models using CT images as input have significantly outperformed the clinical baseline, corroborating the hypothesis of CT images being critical for accurate mortality prediction. Interestingly, the imaging-only model had the best discrimination performance, while the multi-modal model was the best-calibrated model. The imaging model had better discrimination performance compared to the model that used the two sources of information. One reason for this is the noise in clinical data, especially the FVC percent and $D_{LCO}$ features, that impaired the model performance. This supports our hypothesis that depending solely on clinical data for mortality prediction is insufficient. The results suggest that clinical data might not be critical for risk stratification in IPF patients but becomes useful if estimating time to death is the end goal. Additionally, we can see that incorporating a memory bank in the model design consistently improves the predictive performance. Especially in the multi-modal model, the introduction of a memory bank gives an improvement of 1.55% and 0.22 in terms of C-Index and IBS, respectively.

Further, to assess the generalization of our approach, we validate the models on an independent cohort of 107 IPF patients, see Table 2. We use a trained model from our cross-validation experiment and test its performance on this unseen cohort of patients. The

Table 2: Generalization experiments on an independent cohort of patients

| Model | IPCW C-Index | IBS |
|---|---|---|
| Cox (clinical data) | 65.14 | 0.11 |
| CNN (imaging data with memory bank) | 64.05 | 0.49 |
| CNN (both with memory bank) | **67.85** | **0.07** |

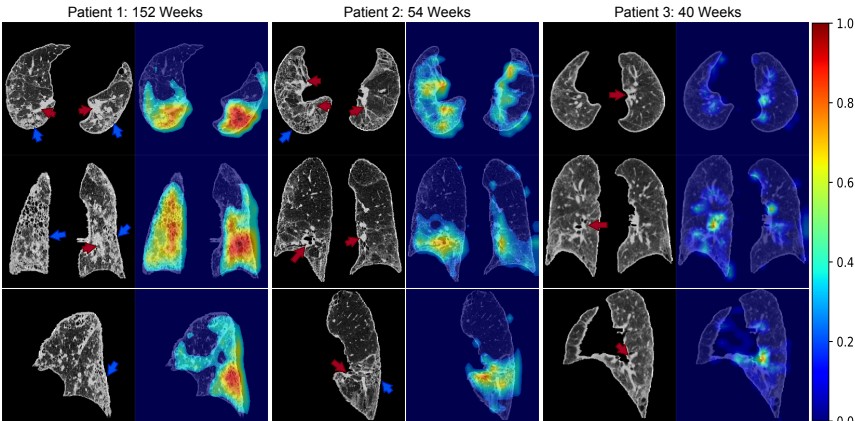

Figure 4: Saliency maps of the survival analysis models with the reported time of death in patients with IPF. The model highlights areas of fibrosis (blue arrows) but also pulmonary vessels (red arrows). Larger samples are presented in Appendix D.

results show that the performance drop in the proposed multi-modal method is the lowest and it has the best generalization in terms of calibration and discrimination performance.

Additionally, we present saliency maps using GradCAM method (Selvaraju et al., 2017) to highlight prognostically important structures on CT images that caused certain predictions by the model (Figure 4). GradCAM computes saliency maps by multiplying the activations of the last convolutional layer by the gradients of the final fully-connected layer, resulting in a low-resolution saliency map which is then upsampled to the original input size. We notice that the model highlights areas of fibrosis and vessels in these patients. This shows that fibrosis extent is a prognostically important imaging biomarker in IPF. Interestingly, the highlighting of vessels confirms the correlation between mortality and pulmonary vessels that has been suggested in IPF, using CALIPER features (Jacob et al., 2017).

## 4. Conclusions and Future Work

We proposed a principled framework to predict survival in IPF from CT images and incomplete clinical data. Our results show that i) the integration between imaging and clinical data gives the best prediction of time to death, while imaging only is sufficient for death risk stratification; ii) using memory banks for approximating Cox loss improves the discrimination and calibration of survival models; iii) pulmonary vessels and fibrosis are prognostically important in IPF. The presented methods could be extended to other modalities and diseases with minor tweaks to adapt the CNN architecture to the modality of interest.

A limitation with GradCAM as an interpretability method is the generation of coarse saliency maps due to the upsampling operation, which leads to highlighting irrelevant areas in some cases. A natural future direction would be designing methods that directly generate high-resolution maps, rather than upsampling low-resolution ones. Further, a comparison between our approach and visual scoring in terms of predictive performance and processing time will demonstrate the practical utility of fully-automated survival analysis methods.

## Acknowledgments

This work is supported by the Open Source Imaging Consortium (OSIC).

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

## Appendix A. Limitations of IPF progression modelling from clinical data

| Age | 69 years | | Age | 73 years |
|---|---|---|---|---|
| Sex | Male | | Sex | Male |
| Smoking History | Ex-smoker | | Smoking History | Ex-smoker |
| Treatment | Antifibrotic | | Treatment | Antifibrotic |
| FVC Percent | 85.0 | | FVC Percent | 86.0 |
| $D_{LCO}$ | 19.14 | | $D_{LCO}$ | 15.08 |
| *Died after 385 weeks* | | | *Died after 114 weeks* | |

Figure 5: An example from the OSIC dataset of two patients with very similar clinical features and different survival outcomes. This illustrates the limitations that exist when only using clinical data to predict disease progression in IPF. Our study examined the additional value that might be gained by using imaging data to predict disease progression. Time of death is reported relative to the time of lung function tests.

## Appendix B. EM algorithm

The EM algorithm maximises the energy term (see Barber (2012)), given a posterior $q(h|x)$:

$$\sum_{n,h} \mathbb{E}_{q(h|x^n)}[\log p(x^n, h)] = \sum_{n,h} \sum_{i \in x^n} \mathbb{E}_{q(h|x^n)}[\log p(x_i|h)] + \sum_{n,h} \mathbb{E}_{q(h|x^n)}[\log p(h)] \quad (9)$$

where $q(h|x^n)$ is given by the E-step:

$$q(h|x^n) \propto p(h)p(x^n|h) \quad (10)$$

An important property of an imputation model is to be able to train using samples with missing values, which can be simply achieved in this model by introducing the set of missing variables $x_m$ as an additional hidden variable and using $x_o$ for the observed variables, in this case the energy will be as follows:

$$\sum_{n,h,x_m^n} \mathbb{E}_{q(h,x_m^n)} \log p(x_m^n, x_o^n, h) = \sum_{n,h,x_m^n} \mathbb{E}_{q(h,x_m^n)}[\log p(h) + \log p(x_m^n|h) + \log p(x_o^n|h)] \quad (11)$$

**E-Step:**

$$q(x_m^n, h|\mathbf{x^n}) \propto p(h) \prod_{i \in x_o^n} p(x_i|h) \prod_{i \in x_m^n} p(x_i|h) \quad (12)$$

In the M-Step, we will also need $q(h|\mathbf{x^n})$:

$$q(h|\mathbf{x^n}) \propto p(h) \prod_{i \in x_o^n} p(x_i|h) \quad (13)$$

**M-Step:** To get $p(x_k|h)$, Energy $E$ can be re-written as

$$E = \sum_{n,h,x_m^n} q(x_m^n, h|\mathbf{x^n})[\sum_{i\in x_o^n} \log p(x_i|h) + \sum_{i\in x_m^n} \log p(x_i|h) + \log p(h)] \tag{14}$$

Since we are interested in optimizing $p(x_i = C|h^n)$:

$$E = \sum_{n,i\in x^n} [\mathbb{I}(x_i = C)q(h|\mathbf{x^n}) \log p(x_i = C|h) \tag{15}$$
$$+ \mathbb{I}(i \in x_m)q(x_i = C, h|\mathbf{x^n}) \log p(x_i = C|h)]$$

where $C$ is a category in the categorical distribution. The first term models the observed values and the second term models the missing values.

$$p(x_i = C|h) \propto \sum_n [\mathbb{I}(x_i^n = C)q(h|\mathbf{x^n}) + \mathbb{I}(i \in x_m)q(x_i^n = C, h|\mathbf{x^n})] \tag{16}$$

To compute $p(h)$:

$$p(h) \propto \sum_n q(x_m^n, h|\mathbf{x^n})$$
$$= \sum_n q(x_m^n|h, \mathbf{x^n})q(h|\mathbf{x^n}) \tag{17}$$

If no missing values, i.e. $x_m^n = \emptyset$:

$$p(h) = \sum_n q(h|\mathbf{x^n}) \tag{18}$$

E and M steps are repeated until convergence.

## Appendix C. Correlation between clinical variables

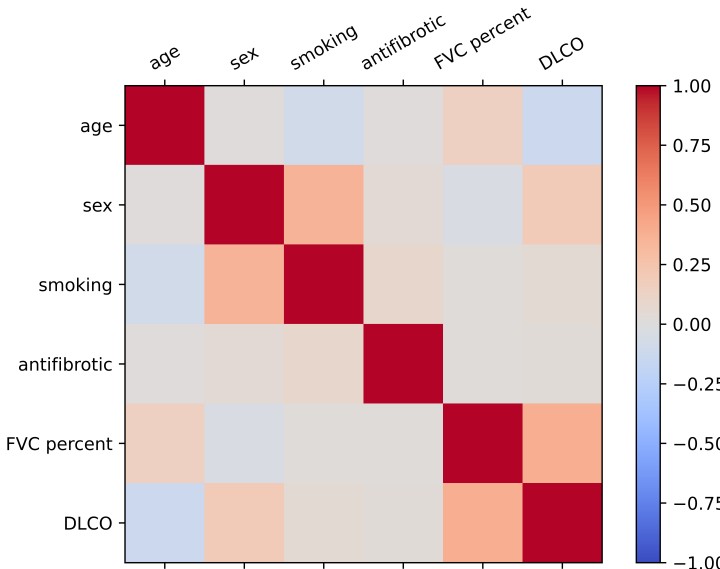

Figure 6: Pearson correlation coefficient between clinical variables. There is some correlation between FVC percent and age while strong correlation between FVC percent and $D_{LCO}$. This illustrates the limitation of methods that assume independence between features.

## Appendix D. Saliency Maps

We show larger samples of the saliency maps with the reported time of death in patients with IPF. The model highlights areas of fibrosis (blue arrows) but also pulmonary vessels (red arrows).

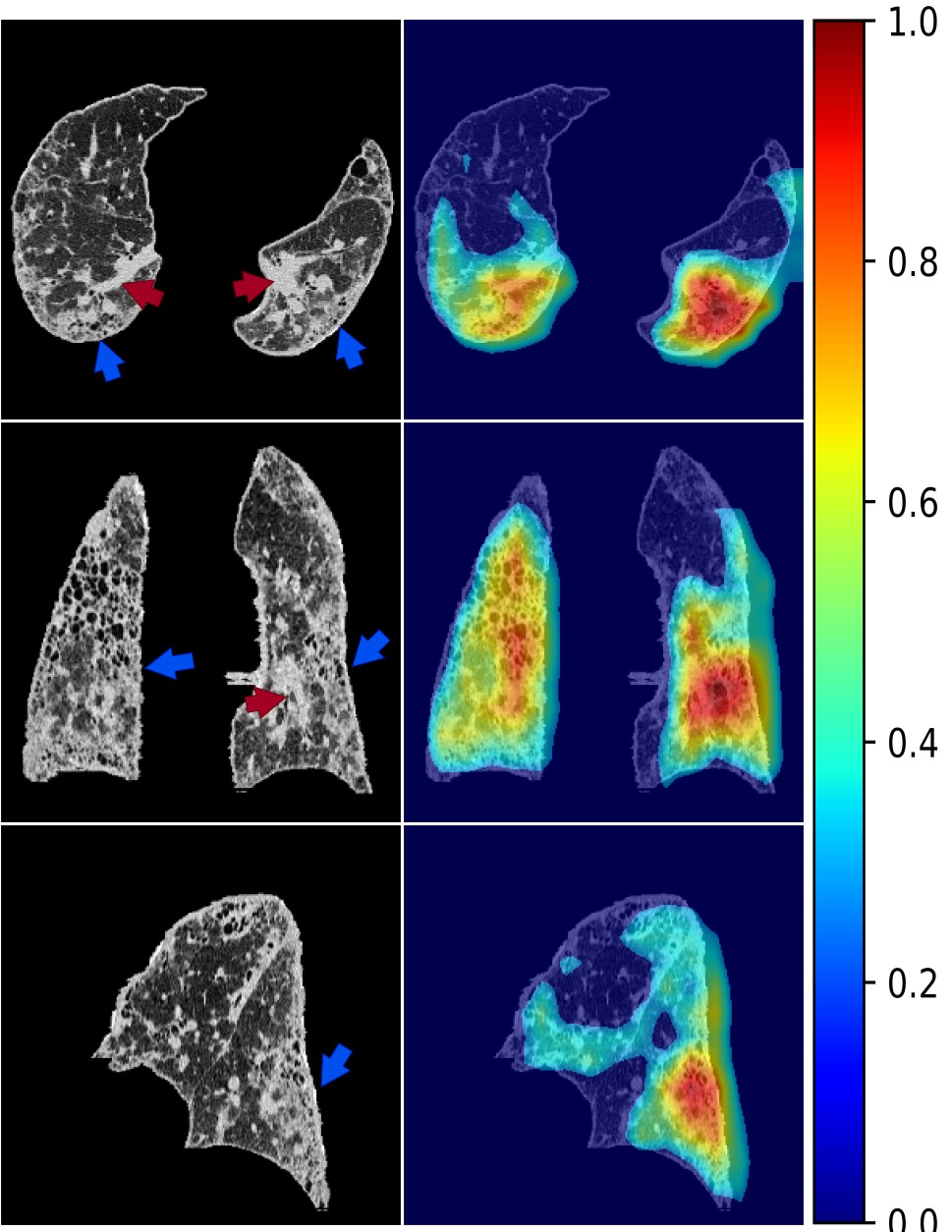

Figure 7: Patient 1. Time of death: 152 weeks.

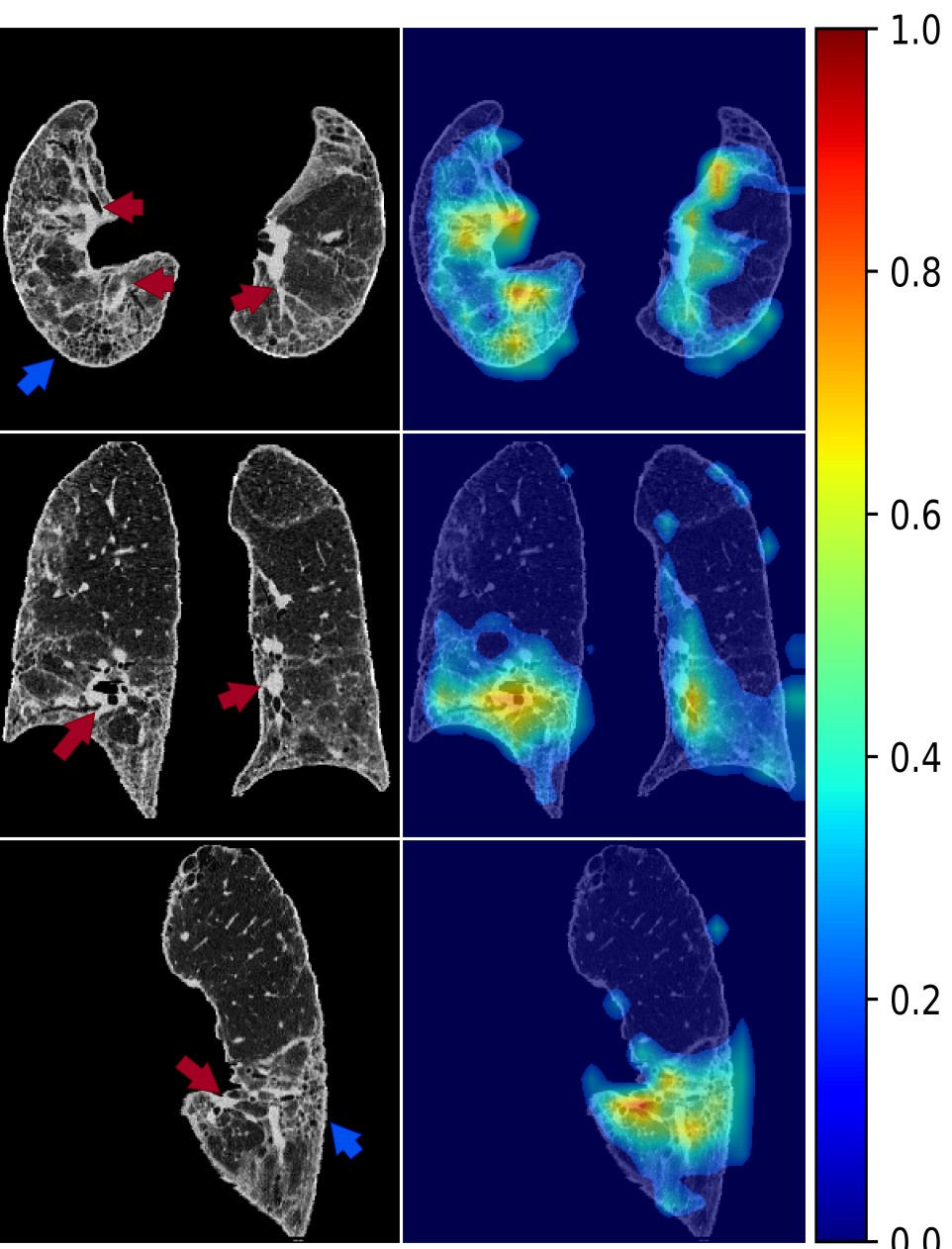

Figure 8: Patient 2. Time of death: 54 weeks.

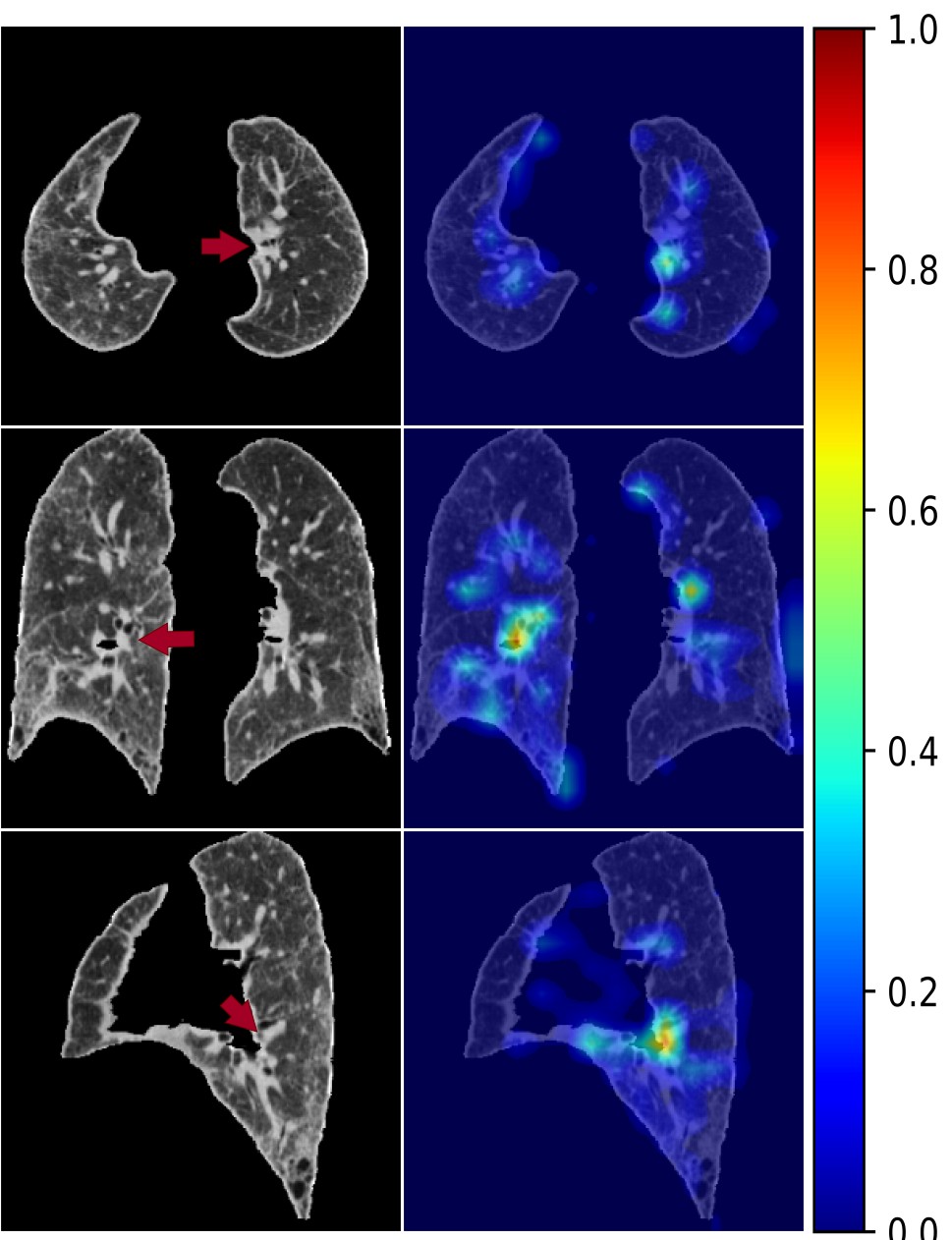

Figure 9: Patient 3. Time of death: 40 weeks.

