# OpenReview forum: "Survival Analysis for Idiopathic Pulmonary Fibrosis using CT Images and Incomplete Clinical Data"
_MIDL.io/2022/Conference — MIDL 2022_

### Official Review · Reviewer_KcKg · 2022-01-12

**Confidence:** 5
**Preliminary Rating:** 5
**Recommendation:** Best Paper Award, Oral

**Summary:**

This work is very interesting. In this work, the author has proposed a multi-modal method that uses neural networks and memory banks to predict the survival of IPF patients using clinical and imaging data. They proposed a probabilistic model that captures the dependencies between the observed clinical variables and imputes missing ones.
This is a very promising way for medical science.


**Strengths:**

This work has been performed very well and well written. The author tried to show a principled survival analysis framework that consists of a simple latent variable imputation model to infer the missing data in clinical records; Very interesting work that is useful in medical science.

**Weaknesses:**

Everything is done in a very proper manner, Abstract is very structured. Equations are well defined, images are of very good quality and references are very relevant. A little weakness is from my side that the author needs to explain a little bit that clinicians can understand.

**Deanonymize Review:**

yes

**Detailed Comments:**

No comments.

**Final Rating After The Rebuttal:**

5: Strong Accept

**Justification Of The Final Rating:**

The paper is well written, and it explains the proposed methodology clearly. The authors addressed prior work nicely, and this proposed method to estimate missing clinical observations provided superior performance.
I think it looks in more shape and has to accept now.
Whatever, we were expecting to change, now it has been done. So I would like to accept this paper.

**Paper Type:**

both

**Questions To Address In The Rebuttal:**

I do think that I need to ask anything from the author. He has shown everything in the paper.
I am just curious to know that it is possible to apply this same concept the other modalities like MRI or Ultrasound.

**Special Issue:**

yes

---

### Meta-Review · Area_Chair_eYnA · 2022-02-18

**Recommendation:** Accept (Oral)
**Confidence:** 5

**Metareview:**

This work is very clear, of very good quality, significant and original. The authors have addressed all the comments of the reviewers which appear clearly in the updated document. I therefore suggest oral acceptance.

I also repeat here the pros I've identified.
- The work uses an original probabilistic model to capture dependencies between observed clinical variables and imputes missing ones.
- Using memory bank in survival analysis has improved the metrics used
- The work adequately addresses and cites previous work.
- The authors demonstrate their approach using three different cohorts of data.
- The results presented in this work are significant and could be applicable to other diseases where survival analysis is performed.

Based on the many pros, the quality of the work and the effective response to reviewers, I propose to retain this work for an oral presentation.

---

### Decision · Program_Chairs · 2022-02-28

Accept